# Hydrogel-Based Skin Regeneration

**DOI:** 10.3390/ijms25041982

**Published:** 2024-02-06

**Authors:** Zohreh Arabpour, Farshad Abedi, Majid Salehi, Seyed Mahbod Baharnoori, Mohammad Soleimani, Ali R. Djalilian

**Affiliations:** 1Department of Ophthalmology and Visual Science, University of Illinois, Chicago, IL 60612, USA; arabpour@uic.edu (Z.A.); fabedi3@uic.edu (F.A.); sbahar2@uic.edu (S.M.B.); msolei2@uic.edu (M.S.); 2Department of Tissue Engineering, School of Medicine, Shahroud University of Medical Sciences, Shahroud 3614773955, Iran; msalehi.te1392@gmail.com

**Keywords:** hydrogel, wound healing, tissue engineering, biopolymer, skin regeneration

## Abstract

The skin is subject to damage from the surrounding environment. The repair of skin wounds can be very challenging due to several factors such as severe injuries, concomitant infections, or comorbidities such as diabetes. Different drugs and wound dressings have been used to treat skin wounds. Tissue engineering, a novel therapeutic approach, revolutionized the treatment and regeneration of challenging tissue damage. This field includes the use of synthetic and natural biomaterials that support the growth of tissues or organs outside the body. Accordingly, the demand for polymer-based therapeutic strategies for skin tissue defects is significantly increasing. Among the various 3D scaffolds used in tissue engineering, hydrogel scaffolds have gained special significance due to their unique properties such as natural mimicry of the extracellular matrix (ECM), moisture retention, porosity, biocompatibility, biodegradability, and biocompatibility properties. First, this article delineates the process of wound healing and conventional methods of treating wounds. It then presents an examination of the structure and manufacturing methods of hydrogels, followed by an analysis of their crucial characteristics in healing skin wounds and the most recent advancements in using hydrogel dressings for this purpose. Finally, it discusses the potential future advancements in hydrogel materials within the realm of wound healing.

## 1. Introduction

The skin, as the largest organ of the human body, consists of three interconnected layers—epidermis, dermis, and hypodermis—which are sequentially placed from the outermost layer to the innermost layer [1].

Every year, globally, numerous individuals undergo invasive procedures to address skin tissue defects. Despite various approaches supporting skin wound healing, none have successfully replicated the microenvironment of the extracellular matrix. Tissue engineering has recently emerged, offering the potential to regenerate tissue defects that the body struggles to restore [2]. This innovative technique in the medical field ensures damaged tissue recovery and utilizes diverse biomaterials to facilitate the growth of body tissues and organs. Synthetic and natural biomaterials play a pivotal role by supporting the extracellular matrix and regenerative capacities in tissue and organ growth [3].

Notably, recent tissue engineering efforts have highlighted the regeneration of skin tissue’s significance, demonstrating promising results in clinical studies for diabetic and burn wound patients. In recent years, the demand for polymer-based therapeutic approaches in skin tissue defects has risen substantially. Contemporary polymer research now focuses on constructing scaffolds using polymeric materials that integrate with biological molecules or cells, promoting tissue regeneration [4].

For the regeneration of skin wounds, diverse polymer-based wound dressings have been developed, including films, gauze, foams, nanofibers, and hydrogels [5]. Hydrogels specifically constitute three-dimensional interconnected networks constructed from hydrophilic polymers [6]. These unique properties, including their special structure and high-water content, confer hydrogels with exceptional oxygen permeability and the ability to absorb wound exudate effectively. Additionally, they create a moist environment at the wound interface. Thanks to their adaptable physical and chemical characteristics, hydrogels can mimic the composition and mechanical attributes of natural tissues, offering sufficient space and mechanical support for cell migration and tissue regeneration [7]. Hydrogels can also be designed for therapeutic agent delivery, responsiveness to specific stimuli, and intelligent monitoring. They can act as carriers for therapeutic substances like drugs, cells, and nanoparticles. Stimuli-responsive hydrogels can serve as intelligent materials that release loaded active substances on demand [8]. Furthermore, hydrogels with monitoring capabilities can continuously assess the wound site’s microenvironment, providing valuable data for treatment planning [9].

This article begins by providing an overview of the anatomy and function of skin, the wound healing process, and traditional wound dressings; then, it focuses on polymers utilized in novel hydrogel scaffold fabrication for skin tissue engineering and, subsequently, the methods used to create them. Finally, this paper outlines the prospects of the application of hydrogels in wound healing.

## 2. Wound Healing Process

Disruption of the structural integrity of the skin causes ulcers. Depending on the extent of tissue damage, wounds are either superficial, which are minor epidermal injuries that affect the outermost layer of the epidermis, or deep, which include damage to the dermis and subcutaneous layers [10].

Upon the occurrence of a wound, the human body initiates a rapid healing response, comprising a complex process divided into four dynamic and overlapping stages. These stages involve the interaction of various cell types and matrix components, including hemostasis, inflammation, proliferation, and remodeling, as illustrated in Figure 1 [11]. In the hemostasis stage, platelets aggregate around damaged blood vessel walls to create a platelet plug, while fibrin forms a blood clot at the wound site [12]. The inflammation stage is prompted by growth factors released by platelets, during which neutrophils and macrophages combat bacteria and break down necrotic tissue [13]. The proliferation stage encompasses extracellular matrix (ECM) deposition, neovascularization, the formation of granulation tissue, and epithelial generation, ultimately resulting in full wound closure. In the matrix remodeling phase, granulation tissue gradually evolves into scar tissue with diminished blood vessels during the final stage of wound healing. In most cases, the mechanical properties of the healed wound site are affected, and regenerating skin appendages is a challenging process [14].

## 3. Approaches for Skin Wounds

In cases of skin injuries, effective wound treatment is very important to prevent bacterial infection and accelerate wound healing. In non-healed wounds such as diabetic wounds, the initial healing phase is disrupted and, due to the failure of the healing process, it progresses to chronicity. In large burn wounds, they prevent normal healing due to significant loss of skin tissue or inability to form a temporary ECM matrix due to tissue necrosis [15]. Therefore, clinical interventions are necessary to treat these wounds. The most effective approach involves providing an artificial matrix, such as a wound dressing or skin graft, which acts as a temporary matrix and aids the healing process. Researchers have tirelessly developed various types of wound dressing materials or skin substitutes using advanced technologies to simulate the microenvironment of healing wounds [16]. The natural wound environment consists of an ECM platform with cell adhesion sites, guiding biochemical signals, growth factors, and different cell types. However, replicating this entire microenvironment is very challenging, leading most tissue engineers to focus on the critical structural framework involved in the healing process [17].

Significant progress has been made in the development of bioactive formulations and regenerative matrices as they initiate the essential steps of the natural healing process and help regenerate the skin’s functional tissue. In general, the use of wound dressings or skin graft platforms helps the wound heal faster and complements the natural healing process. Consequently, the development of a new generation of interactive dressings that combine physical protection with the capacity to accelerate wound tissue regeneration is essential for the treatment of skin wounds [18].

## 4. Recent Strategies for Wound Healing and Skin Regeneration

The wound healing process aims to swiftly and efficiently regenerate skin tissue while minimizing scarring and preventing keloids. However, achieving successful skin tissue repair poses a considerable challenge in biology and healthcare. Persistent wounds, like those seen in diabetic or ischemic cases, often result in reduced function, heightened pain, and increased risk of infection. To address this, innovative and efficient treatments are undergoing intensive development to promote complete skin tissue repair while minimizing complications and costs [19].

In recent years, the focus of research, especially in tissue engineering, has heavily emphasized the creation of structures that enhance reparative properties in wound healing and regeneration. The objective here is to engineer scaffolds that guide cellular behavior to effectively mend damaged skin. Cells not only require a supportive structure to inhabit and multiply but also depend on signals to navigate toward the optimal environment for regrowth. These signals involve intricate interactions of signaling molecules and the physical characteristics of the scaffolds [20,21]. Functionalized scaffolds are integrated with bioactive substances such as growth factors, antimicrobial agents, bioactive nanoparticles or liposomes, cell-binding peptides, and other specific additives to facilitate chemical communication [3].

## 5. Hydrogels for Wound Healing

In addition to covering the wound, the new dressings also help to speed up the healing process of the skin. A significant advancement in this field revolves around the recent emergence and widespread attention given to hydrogels [22]. Hydrogels are three-dimensional polymeric networks with a hydrophilic nature, capable of retaining a substantial amount of water within their structure while retaining their form. These hydrogels possess biocompatibility, undergo biodegradation, and demonstrate favorable attributes. These include elasticity akin to natural tissue; maintenance of a moist environment; absorption of wound exudates; and a porous structure facilitating gas exchange, acting as a barrier against bacterial infections, and creating a conducive setting for enhancing cellular functions like migration and proliferation. Moreover, hydrogels aid in wound hydration, fostering a moist milieu that supports the removal of wound debris through autolysis [23].

## 6. Polymer Selection for Hydrogel

Wound healing involves a complex interplay of external factors that significantly impact its outcome. When selecting materials for wound treatment, specific properties become crucial. Among these, one of the foremost external factors influencing wound healing is the biocompatibility of the materials used in wound dressings. Effective wound dressing materials should regulate moisture levels around the wound, eliminate excess secretions, shield the wound from infections and microbes, minimize wound surface necrosis, and offer mechanical protection [24].

In recent years, there has been a rapid expansion in skin tissue engineering, primarily focusing on the development of hydrogels using diverse polymeric materials. These hydrogels can be categorized into four types of natural, synthetic, hybrid, and biomimetic based on the origin of their components [25].

## 7. Natural Polymers

This group encompasses a range of polysaccharides and proteins predominantly sourced from animals, microorganisms, and plants. In medical contexts, these natural polymers exhibit superior biocompatibility, biodegradability, and/or biological activity compared to their synthetic counterparts (Table 1). However, natural polymers usually show less mechanical strength; to solve this defect, they need modifications such as grafting, cross-linking, and combination with other polymers. Another significant aspect of natural biopolymers is that upon enzymatic degradation, they generate byproducts that are generally well-tolerated by living organisms and do not trigger toxic reactions [26].

### 7.1. Collagen

Collagen, a fundamental protein within the extracellular matrix, is widely distributed throughout mammalian tissues like skin, cartilage, tendons, ligaments, and others, contributing approximately 25–33% to the body’s total protein content [27]. Its extensive use in biomaterials stems from its remarkable biological functions, including fostering cell adhesion and proliferation, as well as its biocompatibility, low immunogenicity, swift hemostasis, and biodegradability. Among collagen types, type I is the most prevalent and broadly dispersed, constituting roughly 90% of the total collagen in living organisms, and it is also the most extensively studied and utilized [28]. Gaspar et al. [29] devised 3D porous dressings using type I collagen extracted from gilthead bream skin, illustrating their ability to effectively enhance fibroblast migration and consequently promote wound healing. Ying et al. [30] developed an injectable hydrogel dressing based on collagen, which notably expedites the formation of blood vessels, epithelization, collagen deposition, and the natural process of wound healing.

**Table 1 ijms-25-01982-t001:** Advantages and disadvantages of the discussed natural and synthetic polymers.

Polymer Type	Disadvantages	Advantages	Ref.
Natural polymers	Poor Mechanical Strength	Biocompatibility	[31,32]
	Rapid Degradation		
	Inflammatory Response	Biodegradability	
	Potential Allergic Reactions	Cell Affinity	
	Sourcing Challenges	Moisture Retention	
	Limited Control over Properties	Angiogenic Properties	
	Pathogen Contamination		
Synthetic polymers	Biocompatibility Concerns	Tunable Properties	[33,34]
		Tailored Biodegradability	
	Inflammatory Responses	Consistent Composition	
	Lack of Bioactivity	Mechanical Strength	
	Potential Toxicity	Absorption Capacity	
	Limited Water Retention	Pore Size Control	
	Mechanical Weakness	Long Shelf Life	
	Non-Natural Structure	High degree of sterility	
		Customized Drug Delivery	

### 7.2. Chitosan

Chitosan is a type of polysaccharide that is widely used for various biomedical applications. Through partial deacetylation of chitin, which is the second most abundant polysaccharide found in the exoskeleton of crustaceans and insects and in the cell wall of fungi, it is obtained by two chemical or biological methods [35]. Chitosan has a number of beneficial properties including biocompatibility, biodegradability, and antimicrobial activity. These properties make chitosan a very attractive biomaterial for use in a number of biomedical applications, including tissue engineering, drug delivery, vaccine administration, and medical device manufacturing [36]. In addition, chitosan can be converted into various formulations such as nanoparticles, nanofibers, scaffolds, hydrogels, membranes, films, and many others, making it suitable for specific biomedical applications. Chitosan can be chemically and physically modified to provide tailor-made formulations [37]. In order to improve the mechanical properties and bioactivity of hydrogel dressings, Wu et al. [38] developed a chitosan/alginate hydrogel dressing with bioglass (BG) as a bioactive ingredient. Calcium ions released from BG led to increased alginate binding and mechanical stability of hydrogels. Upon contact with wound exudate, the designed dressing was able to protect the skin wounds against bacterial infections by maintaining the proper moisture of the healing environment. Carboxymethyl chitosan (CCS), derived from chitosan, is soluble in water and demonstrates remarkable water retention properties [39]. Cao et al. [40] utilized collagen and CCS as primary ingredients to create a hydrogel resembling the structure and makeup of the human extracellular matrix, providing nutrients for cell growth and proliferation.

### 7.3. Hyaluronic Acid (HA)

Hyaluronic acid (HA) was initially identified in 1934 and has undergone extensive investigation since then [41]. HA, a natural polysaccharide, constitutes a vital component within the extracellular matrix (ECM) and consists of a linear polysaccharide featuring alternating units of b-1,4-linked D-glucuronic acid and (b-1,3) N-acetyl-D-glucosamine4. HA showcases a range of physiological properties, encompassing structural support, space-filling capacity, lubrication, absorption, and retention of water in tissues and the ECM [42]. HA serves as a biocompatible polymer with multiple biological functions, such as moisturizing the skin and exhibiting anti-wrinkle effects that facilitate the natural healing process. Moreover, numerous studies conducted in vitro and in vivo have highlighted HA’s role in wound healing by enhancing the migration and differentiation of mesenchymal and epithelial cells, as well as fostering angiogenesis and collagen deposition. Furthermore, HA fosters an advantageous microenvironment for wound healing, exhibiting significant potential in the treatment of scarring and overall wound recovery [43].

De Fransisco et al. [42] showed that topical application of 0.2% hyaluronic acid as a topical application can improve healing by encouraging matrix formation and reducing inflammation, achieved through maintaining a moist wound environment. Also, this moisture-rich environment was able to help prevent tissue dehydration and cell loss, accelerate the formation of new blood vessels, accelerate the breakdown of dead tissue and fibrin, increase cell interaction, and in particular, reduce pain.

Hu et al. [44] formulated a wound dressing through the electrostatic self-assembly of hyaluronic acid and ε-polysine. This injectable dressing demonstrated the ability of self-healing along with antibacterial effects. They showed that due to its exceptional adhesion and strong antibacterial function, it effectively eradicates bacteria, reduces inflammation, encourages collagen formation, promotes keratinocyte motility, facilitates angiogenesis. It slows down and accelerates the wound healing process.

### 7.4. Gelatin

Gelatin is a natural protein derived from collagen hydrolysis sourced from the connective tissues of animals like skin, bone, and tendons. Throughout history, gelatin has served as both an antipyretic and hemostatic drug and has been utilized as a component in wound dressings [45]. Its richness in arginine–glycine–aspartic acid (RGD) sequences makes gelatin advantageous for cell attachment and regeneration without necessitating RGD cross-linking, setting it apart from other biomacromolecule-based hydrogels. Furthermore, recent attention has been directed toward gelatin-based hydrogels in biomedical applications due to their biocompatibility, biodegradability, and adaptability for chemical modifications [46].

Xu et al. [47] designed a 3D bioprinted hydrogel patch using UV-crosslinked methacrylated gelatin (GelMA) and silk fibroin (SilMA) to accelerate acute wound healing. The combination of GelMA with silk fibroin (GelSilMA) led to improved biodegradability and mechanical properties. In addition, the designed hydrogel was able to maintain a moist healing environment in the wound area with sufficient degradation capacity. Also, GelSilMA (G-S) hydrogel was able to accelerate wound closure by improving the microenvironment for epidermal tissue regeneration and endogenous collagen production.

### 7.5. Alginate

Alginate, a hydrophilic and anionic polysaccharide, has demonstrated its worth through several beneficial attributes, including its role in forming gels, its capacity for absorbing proteins, its compatibility with living tissues, its ability to retain moisture, its capability to break down naturally, and its possessing viscoelastic properties. Furthermore, alginate hydrogels are regarded as highly suitable materials for contemporary wound dressings. Wound dressings based on alginate hydrogels come in diverse forms like nanofibers, foams, membranes, and reinforced composites made from non-woven materials [48].

Wang et al. [49] developed a dual-network hydrogel made from sodium alginate and platelet-rich plasma (PRP) to aid in the process of wound recovery. The hydrogel was created by activating thrombin. When topically applied to the skin wounds of mice, the results revealed that this hydrogel, formulated with optimal levels of epidermal growth factor (EGF) and vascular endothelial growth factor (VEGF), could promote cell proliferation, revitalize blood vessel formation, and facilitate the closure of wounds. In the context of wound healing, which involves a highly intricate biological process leading to the formation of fibrotic tissue in the absence of hormonal activity, especially in cases of skin tissue defects resulting from trauma and burns, Wang at al. [50] have developed a sodium alginate hydrogel loaded with Benlysta. This innovative hydrogel exhibits remarkable anti-inflammatory and skin tissue regeneration capabilities. Furthermore, experiments involving the gelation of Benlysta-loaded sodium alginate gels have shown that the gelation time can be controlled by adjusting the sodium alginate ratio, hinting at its potential utility as both a skin dressing and for subcutaneous injection. Moreover, characterization studies have unveiled the hydrogel’s excellent swelling rate, sustained drug release, and biodegradability, as well as its ability to promote fibroblast and epidermal cell proliferation. Ultimately, this Benlysta-loaded sodium alginate hydrogel offers new insights into the treatment and management of skin tissue defects [50].

### 7.6. Dextran

Dextran has played a pivotal role in tissue regeneration research due to its remarkable biocompatibility, biodegradability, and low toxicity. Its optimal release characteristics have led to its utilization in hydrogel scaffold development. In this context, Du et al. [51] showcased the hydrophobic interaction between hydrophobic aliphatic chains in a newly engineered modified chitosan (hmCS) and oxidized dextran hydrogel. Comprehensive characterization studies, rheology, cytotoxicity, and tissue adhesive assessments unveiled these hydrogels’ ideal morphology, viscoelasticity, non-toxicity, and bioadhesive properties. Furthermore, antibacterial investigations demonstrated an impressive bactericidal efficacy of 95.0% and 96.4% at a total bacterial concentration of 108 CFU/mL. Additionally, observations in a wound healing model confirmed the exceptional regenerative potential of these hydrogels in repairing skin tissue defects, opening possibilities for their application in other tissue regeneration approaches [24].

Traditional skin tissue grafts and transplants for burn wound healing often face challenges, including inadequate re-vascularization and re-epithelialization. Synthetic therapies aimed at regenerative wound healing offer a non-immunogenic, off-the-shelf approach to improving the clinical management of burn-related skin tissue defects [4]. Shen et al. illustrated the effective mechanisms of dextran-based hydrogel in burnt wound healing. Their model was designed to analyze the clinical applicability of hydrogel therapy and its role in tissue regeneration. The initial anti-inflammatory response of the dextran-based hydrogel stimulated angiogenesis, followed by successful re-epithelialization, fostering an effective skin tissue regeneration process. This advancement in dextran-based hydrogel technology has paved the way for clinical trials aimed at enhancing the treatment of patients with severe burns [52].

Burn wounds carry significant risks, including infection and pathogenic scar tissue formation, which contribute to delays in wound closure and an increase in wound-related complications. To address this, Zhu et al. [40] developed a dextran-hyaluronic acid (Dex-HA) hydrogel incorporating sanguinarine (SA) gelatin microspheres (GMs). This formulation exhibited ideal porosity (80%) and a notable swelling ratio, promoted the proliferation of NIH-3T3 fibroblast cells, and maintained a controlled release profile for SA within the GM. In vitro studies indicated faster degradation and significant antimicrobial potency for the SA GMs-incorporated Dex-HA hydrogel. Moreover, histopathological and immunohistochemical findings supported re-epithelialization and re-vascularization in the hydrogel treated groups, demonstrating the effectiveness of SA GMs-incorporated hydrogel in the regeneration of burnt skin tissue with minimal scar formation [40].

### 7.7. Fibrin

Fibrin, a crucial element in skin regeneration, forms as a biopolymer through enzymatic reactions within the coagulation cascade. When this cascade activates, soluble fibrinogen transforms into an insoluble network of fibrin fibers under thrombin’s influence. This network stabilizes the platelet plug by attaching platelets to the fibrin, prompting the release of growth factors like platelet-derived growth factor. These growth factors stimulate fibroblasts to multiply, migrate to the wound site, and produce essential components such as collagen I and III, glycosaminoglycans, and proteoglycans. Additionally, the fibrin network links with various cell types like smooth muscle cells and endothelial cells via integrin adhesion receptors. Moreover, it interacts with cell adhesion-mediating ECM proteins like fibronectin and vitronectin, facilitating fibroblast adhesion [53].

Xu et al. [54] introduced a pioneering approach by developing innovative granule-lyophilized platelet-rich fibrin-loaded polyvinyl alcohol hydrogel scaffolds as an ideal material for wound dressings in the context of skin tissue regeneration. Notably, comprehensive characterization studies highlighted the exceptional qualities of these scaffolds, including their impressive morphology, biocompatibility, biodegradability, and substantial mechanical strength. Subsequently, in vivo assessments, histopathological examinations, and immunohistochemical analyses demonstrated the scaffolds’ ability to promote optimal granular tissue formation, re-epithelialization, and revascularization, suggesting their potential as ideal dressings for topical wounds [54].

Burmeister et al. [55] developed a fibrin-based hydrogel to expedite tissue regeneration in burn wounds. Subsequent histopathological and histochemical investigations revealed that these fibrin hydrogels enhanced granular tissue formation without impeding the re-epithelialization process. Furthermore, when compared to other skin substitutes, the immunomodulatory effects of PEG-fibrin hydrogels played a significant role in accelerating the healing of burn wounds [55].

Severe burns have often presented challenges related to the accessibility of typical skin tissue sources, leading to fatal outcomes for patients. To address this issue, Natesan et al. [56] devised a collagen–polyethylene glycol (PEG) fibrin-based bilayer hydrogel loaded with debrided skin adipose stem cells (dsASCs) for the treatment of severe burns. The novelty of this concept lies in the isolation of stem cells from the patient’s burnt wound site, which significantly enhances the tissue compatibility of these hydrogels. Finally, both in vitro and in vivo studies provide evidence for the utility of dsASCs-loaded hydrogels in the treatment of skin tissue defects [56].

### 7.8. Silk

Silk has been renowned in textile industries for centuries and has served as sutures in medicine for decades. It is a natural macromolecular protein polymer produced within specialized glands’ epithelial cells, secreted into a lumen, and spun into fibers by various Lepidopteran larvae [57]. These fibers consist of two proteins: fibroin, the central protein; and a surrounding adhesive coating made of sericin. Research has revealed that sericin-free fibroin fibers demonstrate exceptional biocompatibility both in controlled laboratory settings and within living organisms [58]. Recently, silk fibroins (SFs) have garnered increasing attention for biomedical applications due to their favorable biocompatibility, adjustable degradation rates spanning from short hours to several years, and impressive mechanical properties when shaped into diverse forms. Moreover, the capacity to finely adjust the molecular structure and morphology through diverse aqueous or organic solvent-based methods and surface modifications has broadened its potential uses in various biomedical applications, including drug/protein/gene delivery and tissue engineering [59].

Fathi et al. [10] developed a hybrid fiber scaffold containing chitosan, PVA, and silk using an electrospinning technique to assess their combined impact on the wound healing process. The hybrid chitosan–PVA + silk fiber scaffold exhibited notably enhanced mechanical properties and achieved desired swelling characteristics, along with creating a more hydrophilic microenvironment compared to pure PVA and chitosan–PVA fibers. The incorporation of blended chitosan and electrospun silk within the PVA-based fibrous scaffold notably improved cell adhesion and proliferation when compared to both neat PVA and chitosan–PVA fibers. The in vivo investigation revealed that the composite chitosan–PVA + silk fibrous mats, in the presence of MSC-derived keratinocytes, effectively promoted wound healing and facilitated the regeneration of skin tissue [10].

## 8. Synthetic Polymers

Synthetic polymer hydrogels offer distinct advantages in the field of skin tissue engineering due to their manageable molecular weight and physical and chemical properties. In contrast to natural polymers, synthetic polymers boast greater mechanical strength and an adjustable biodegradation rate (Table 1) [60]. During the preparation of synthetic polymers, there might be residues of small molecules, which could pose a risk of biological toxicity and immune rejection when applied to wounds [61]. To mitigate these concerns, researchers are exploring mimicking the natural polymer structure and enhancing polymer functionality when developing new synthetic polymer-based hydrogels [62].

### 8.1. Polyethylene Glycol (PEG)

Polyethylene glycol (PEG) possesses distinctive characteristics, including excellent hydration abilities, environmental inertness with minimal protein absorption, limited cell activation and attachment, non-toxicity, and strong biocompatibility [63]. Its biocompatibility and solubility in water make it an ineffective substance in biological environments with low protein absorption, rendering it a suitable scaffold for loading nanoparticles and drugs. An et al. [64] created a hydrogel that incorporates PEG–thioketal bonds (PEG-TK) sensitive to reactive oxygen species, aiming to enable the transportation of epidermal growth factor.

Being hydrophilic polymers, PEG tends to absorb moisture when in contact with bodily fluids like blood. PEG-modified polymers resist the absorption of proteins and bacteria, potentially through hydration and steric hindrance effects. Conversely, they can enhance wound healing by creating micro-negative pressure, while high absorption also plays a role. Chen et al. [65] demonstrated that modifying polyurethane (PU) foam dressings with polyethylene glycol (PEG) and triethoxysilane (APTES) aid in wound healing. Dressings modified with PEG (PUE) and PEG/APTES (PUESi) were created via self-foaming reactions. Results indicated that PUE and PUESi groups exhibited superior physicochemical properties than the gas and PolyMem groups. Moreover, PUESi demonstrated enhanced abilities to prevent adhesion, increased absorption capacity with flexibility, and accelerated the inflammatory phase while promoting collagen deposition in both diabetic and non-diabetic animal models [65].

### 8.2. Polydopamine (PDA)

Dopamine, a neurotransmitter possessing antioxidant capabilities due to its catechol functional group, undergoes oxidation in basic pH aqueous solutions, leading to polymerization and the creation of polydopamine. Polydopamine (PDA) has gained widespread attention as a coating material for biomaterials, mainly because of its multiple catechols that mimic adhesive proteins. These catechols also contribute to PDA’s antioxidant properties. Recent findings revealed that polyethylene imine (PEI) and dopamine can copolymerize, forming a PEI–PDA conjugate in a single reaction vessel [66]. The functional groups present on PDA chains possess the ability to interact with chemical functionalities found on the skin’s surface. This characteristic grants PDA tissue adhesive properties, making it particularly suitable for creating self-adhesive dressings. Consequently, leveraging the tissue adhesive qualities of PDA has sparked significant interest in the development of adhesive hydrogels within the biomedical field [67].

Chai et al. devised highly flexible and swelling-resistant hydrogels by incorporating poly (thioctic acid) (PTA) with PDA, capable of serving as both wound dressings and strain sensors simultaneously. PTA, chosen as the fundamental matrix, benefits from long-term tissue adhesiveness. Furthermore, its monomer, thioctic acid (TA), holds significant value as a natural antioxidant widely present in animal and plant tissues, functioning as a scavenger for hydroxyl radicals [68]. Antioxidants, recognized for their ability to enhance the wound healing process, have been explored as a therapeutic approach for treating chronic wounds. Dopamine, a widely acknowledged neurotransmitter, possesses antioxidant properties and can undergo polymerization to yield polydopamine (PDA).

Connor et al. [69] demonstrated that within dextran hydrogels, the incorporation of PEI–PDA ethyleneimine–polydopamine copolymer resulted in the formation of dense, dark gels. This outcome was attributed to a higher presence of polydopamine due to dextran integrated into the hydrogel. The antioxidant effectiveness of polydopamine, as well as the structure and resilience of the hydrogel, relied on the ratio of dopamine used. Notably, in this investigation, in vivo, wound healing exhibited substantial cell migration when employing the PEI–PDA-containing hydrogel [69].

### 8.3. Polyacrylamide (PAM)

The exceptional mechanical and biological characteristics of PAM, along with its ability to exhibit collective behaviors, have rendered it invaluable in the design of hydrogel scaffolds for tissue engineering purposes. Wei et al. [70] developed composite hydrogels comprising polyvinyl alcohol (PVA) and polyacrylamide (PAM), and their work shed light on the collective behavior of nano-silica when incorporated into these composite hydrogels. Notably, the results of their research indicated that higher concentrations of nano-silica resulted in increased aggregation. Moreover, increasing the polyacrylamide concentration concurrently improved the consistency within the hydrogel scaffolds loaded with nano-silica.

In another study by Pafiti et al. [71], polyacrylamide-based composite hydrogels were created with entrapped hollow particles. This research demonstrated the polymer’s effectiveness in producing hydrogels with precisely adjustable mechanical properties. Interestingly, all dispersions in this study showed that the percolation of the hollow particle network effectively increased the modulus and ductility as anticipated. Moreover, the ability of polyacrylamide-based hydrogel scaffolds to respond to changes in pH makes them highly advantageous for use in wound healing scenarios.

### 8.4. Polyvinyl Alcohol (PVA)

Polyvinyl alcohol (PVA) is a water-soluble polymer derived from vinyl acetate polymerization and alcoholization, featuring a wealth of hydrophilic groups that grant it exceptional moisture absorption and enable biodegradation into water and carbon dioxide [72]. A recent study in *Nature* demonstrated a groundbreaking use of PVA, replacing human serum albumin (HAS) to nurture hematopoietic stem cell (HSC) growth and maintain their stemness. This resulted in an unprecedented thousand-fold expansion of HSCs in vitro, highlighting the vast potential of synthetic polymers like PVA in biomedical contexts [73]. PVA has emerged as a promising biomedical material for skin wound healing due to its superior biodegradability, biocompatibility, and non-toxic nature. For instance, PVA sponges act as clinical hemostats in various surgeries, mitigating post-operative bleeding risks [74]. Moreover, diverse physical, chemical, or radiation methods can cross-link PVA to create hydrogel dressings [75]. However, the accumulation of hydrophilic hydroxyl groups in PVA’s molecular chain results in a thicker water layer at secretion sites, potentially leading to bacterial infections. This diminishes its efficacy in treating skin lesions, prompting the exploration of PVA modification using natural polymers or bioactive compounds to address these shortcomings [72].

Fathi et al. [10] devised an electrospun hybrid scaffold composed of polyvinyl alcohol (PVA), chitosan (Ch), and silk fibers. They assessed its wound healing potential via in vitro implantation of keratinocytes derived from bone marrow mesenchymal stem cells (MSCs) and further evaluated it in vivo. Histological analysis and skin regeneration outcomes showcased the potential of this novel fibrous structure as a skin substitute for repairing damaged skin in regenerative medicine applications.

### 8.5. Carboxymethyl Cellulose (CMC)

CMC has become a crucial component in addressing the challenges of cutaneous wound healing worldwide. Bilayered hydrogels have a significant role in designing the process of cutaneous wound healing [76]. The exceptional swelling and mechanical properties of carboxymethyl cellulose make it a valuable material for developing bilayered hydrogel scaffolds intended for skin tissue engineering applications. For instance, Li et al. [77] developed PVA/CMC/PEG hydrogels with varying pore sizes for use in cutaneous wound dressings. Their study findings indicated that these bilayered hydrogel scaffolds, in addition to demonstrating ideal mechanical properties, effectively prevented bacterial penetration into the hydrogel structure. This led to a reduction in wound closure time in animals treated with these scaffolds. Moreover, MTT assays confirmed the excellent biocompatibility of the single-layer hydrogels. Due to its exceptional biocompatibility, biodegradability, and drug entrapment capabilities, cellulose derivatives have proven useful in designing hydrogels for localized treatment of damaged skin areas. Ali et al. [78] developed biofilm-type wound dressings by incorporating graphene oxide into sodium carboxymethyl cellulose hydrogels. These hydrogels facilitated wound healing with impressive antibacterial activity by regulating epidermal and fibroblast growth factors. The study clearly demonstrated that reduced graphene oxide hydrogel played a significant role in countering the virulent factors produced by bacteria [78].

Carboxymethyl cellulose is known for its environmentally friendly characteristics, making it a popular choice for the development of skin repair substitutes. Capanema et al. [79] created potential wound healing hydrogels with outstanding super-absorbent properties by utilizing carboxymethyl cellulose and PEG polymers. Furthermore, their study highlighted the essential role of carboxymethyl cellulose in enhancing super-absorbent properties in these hydrogel dressings.

### 8.6. Carbopol

Carbopol, owing to its impressive gelling properties, has found versatile applications in various pharmaceutical contexts. Furthermore, its positive impact on the stasis zone has rendered Carbopol an ideal polymer for the development of hydrogels in tissue engineering applications [57]. For instance, Hayati et ai. formulated carbomer 940 hydrogels specifically designed for the healing of burn wounds. Their histopathological analysis demonstrated that these carbomer 940 hydrogels significantly enhanced collagen deposition and reepithelization in damaged areas compared to a control group. Additionally, the outcomes from cell viability tests demonstrated that these hydrogels did not exhibit toxicity towards fibroblast cells [58]. Chronic wound treatment has become a pressing global healthcare concern. Leveraging its excellent biocompatibility and soothing properties, Carbopol has been employed to expedite the healing process. According to Grip et al. [59], sprayable Carbopol hydrogel loaded with beta-1,3/1,6-glucan promotes remarkable revascularization and reepithelization in damaged tissues. While Carbopol 971P, known for its exceptional gelling and high transparency characteristics, was preferred in this study, it is important to note that these qualities alone cannot solely reduce the duration of wound healing. In a study by Zinov’ev et al. [60], Carbopol hydrogels were developed for wound healing in an alloxan-induced diabetes rat model. These Carbopol hydrogels were modified using electrical signals and antibiotics like Poviagrol, resulting in enhanced reepithelization and a reduced rate of suppuration. Additionally, this modified Carbopol hydrogel exhibited bactericidal properties, making it a valuable asset in the treatment of necrotic lesions.

## 9. Hybrid Hydrogels

Hybrid hydrogels, formed by combining the advantages of natural and synthetic polymers, provide an opportunity to incorporate additional properties through the inclusion of active nanofillers. In the field of skin tissue engineering, hybrid hydrogels have quickly gained significant attention and momentum in recent years [80]. Qian et al. [61] introduced an effective method of creating a hybrid hydrogel, addressing bacterial infections and hypoxic conditions in persistent diabetic wounds. This hybrid gel, named SF/Hb/Ga, combines silk fibroin networks cross-linked with hemoglobin and gallium. In laboratory tests, this hydrogel notably boosted cell growth. Hemoglobin, acting as a peroxidase, converted H_2_O_2_ into O2, fostering a conducive natural environment for diabetic wound recovery. In animal trials, the SF/Hb/Ga gel expedited all phases of tissue repair, resolving infected diabetic wounds in just 15 days.

To enhance both the mechanical strength and antibacterial features of hydrogels, Guo et al. [62] combined aramid fibers (ANFs) and tannic acid (TA) with PVA hydrogels through multiple freezing-thawing cycles.

Feng et al. [63] created a hybrid hydrogel to enhance the healing of full-thickness wounds, containing a combination of conjugated collagen (AC), oxidized sodium alginate (OSA), and antimicrobial peptides (polymyxin B sulfate and bacitracin). AC, derived from marine fish scales, possesses low immunogenicity and excellent biocompatibility, making it environmentally friendly, cost-effective, and sustainable. This hybrid hydrogel effectively fought against Escherichia coli and Staphylococcus aureus, promoted cell growth and blood vessel formation in laboratory settings, and expedited the healing of full-thickness wounds in mice by accelerating the processes of skin re-growth, collagen buildup, and blood vessel formation. Lei et al. [64] integrated collagen similar to that found in humans along with tannic acid into a PVA/borax mixture, enhancing the clotting abilities and antioxidant traits of the PVA hydrogel. This combination accelerated the process of wound healing. Dou et al. [65] used poly (γ-glutamic acid) (γ-PGA) and gelatin to prepare hydrogel. This combination shows a significant ability to absorb body fluids and accelerate wound closure. Additionally, Shi et al. [66] introduced a novel composite hydrogel (γ-PGA/SS) composed of γ-PGA and silk sericin (SS) to stimulate cell proliferation and regenerate damaged skin tissues. The incorporation of SS not only enhanced its mechanical properties but also expedited the wound healing process.

## 10. Biomimetic Hydrogels 

Hydrogels are made using polymeric materials that have a similar chemical composition, structure, or function to the natural extracellular matrix (ECM). Biomimetic polymer hydrogels not only reflect the structure and composition of the ECM, but also mimic its biochemical functions [67].

The decellularized extracellular matrix (dECM) refers to a matrix substance acquired by eliminating cellular elements from an organ or tissue. This dECM preserves the structural integrity of the ECM and a considerable amount of bioactive molecules, enabling it to facilitate cell attachment, movement, and the regeneration of tissues [68]. Hence, dECM becomes an optimal substance for creating biomimetic polymer hydrogels. For example, Xu et al. [69] developed a bioactive hydrogel by combining dECM with gelatin and chitosan (dECM/Gel/CS). GAGs, extracted from dECM, act as influential regulators crucial in the skin’s wound healing process. Wang et al. observed that GAGs in hydrogels could alleviate inflammation and encourage angiogenesis. Additionally, dECM can function as a carrier for transporting cells, growth factors, and nanoparticles. Bankoti et al. [70] introduced the dissolved decellularized dermal matrix into a chitosan solution modified with carbon nanodots to form a hydrogel. The incorporation of this matrix gave the hydrogel strong bioactivity and angiogenic potential. Although many components of biomimetic polymer hydrogels are derived from the decellularized extracellular matrix, the exact function of different bioactive compounds in dECM in the skin repair process has not yet been fully elucidated. Xia et al. [71] utilized an enhanced Fryetes’ method [72] to create an injectable extracellular matrix hydrogel (dECMH) sourced from human umbilical cord tissue, specifically tailored for treating persistent or resistant diabetic wounds. This method preserved numerous active elements within the extracellular matrix. Both in laboratory settings and animal studies, the dECMH demonstrated outstanding compatibility with biological systems. Additionally, human umbilical vein endothelial cells exhibited remarkably high migration and proliferation rates within these hydrogels. The pre-gel exhibited favorable injectability and transitioned into a hydrogel state within half an hour. Biochemical analysis revealed that the decellularized extracellular matrix retained abundant structural proteins and growth factors crucial for wound healing. Nanofibers produced through electrospinning offer several advantages. They possess a unique surface-to-volume ratio that enhances cell contact, and their structure closely resembles that of the ECM in the skin’s dermal layer [71].

Chen et al. [73] created a photo-crosslinkable composite nanofibrous hydrogel film using methacrylated polyvinyl alcohol/maleilated hyaluronate (MaPVA/MHA). Hyaluronic acid, a primary component of the extracellular matrix, has super absorbent properties that can modulate inflammatory responses and reduce scarring. Nanofibrous structures can foster cell adhesion and migration. The biomimetic MaPVA/MHA nanofiber hydrogel film promoted the growth of both the dermal and epidermal layers of the skin.

## 11. Preparation of Hydrogels

Hydrogels employed in wound treatment can be created through either physical cross-linking or chemical cross-linking methods. The choice of cross-linking method significantly impacts the hydrogel’s gelation rate. Hydrogels with versatile properties can be tailored to cater to various requirements for different types of skin wounds [74].

Physically cross-linked hydrogels frequently demonstrate sensitivity to surrounding conditions and can self-repair. However, their mechanical properties are relatively low, and they suffer from poor stability, which limits their clinical utility [75]. To enhance their mechanical attributes, a combination of different cross-linking methods is often employed [81]. On the other hand, chemically cross-linked hydrogels are known for their strong mechanical attributes and exceptional stability. However, the utilization of initiators and cross-linking agents in chemical cross-linking may increase cytotoxicity and reduce the biocompatibility of these hydrogels [82].

### 11.1. Physical Cross-Linking

Physical cross-linking occurs under mild environmental conditions and arises from physical forces like hydrogen bonds and polar bonds. Hydrogels formed through physical cross-linking generally demonstrate excellent biocompatibility and degradability because these interactions are reversible, and they do not involve toxic chemical cross-linking agents. Physical cross-linking relies on reversible interactions, primarily ionic interactions, hydrogen bonding, self-assembly, and similar mechanisms [83].

Ionic interaction: Ionic interaction involves interactions between polyelectrolytes and oppositely charged substances. However, the mechanical characteristics of these hydrogels are relatively weak, and their stability in physiological settings is restricted. Additionally, the release of ions during the degradation of ionic cross-linked hydrogels might prompt biotoxicity. Selecting the type and concentration of ions is crucial. For example, through ionic interaction, the negatively charged –COO− within alginate molecules can bind with Ca^2+^ and Ba^2+^ ions, resulting in the formation of cross-linked hydrogels [84].

Rezvanian et al. [85] developed hydrogel films of alginate–pectin for wound healing, which are linked to simvastatin through ionic cross-linking to enhance mechanical strength, wound fluid absorption, and drug release properties. Alginate–pectin hydrocolloid films were chemically cross-linked by immersion in different concentrations of CaCl_2_ (0.5–3% *w*/*v*) for 2 min. This cross-linking significantly improved both the mechanical properties and the absorption capacity of the wound fluid. Furthermore, these hydrogel films maintained their physical structure during use, with the most favorable results observed in films with lower levels of cross-linking. Thermal analysis confirmed that the cross-linking process increased the thermal flexibility of the hydrogel films.

Hydrogen bonding refers to interactions between hydrogen atoms and electronegative atoms such as nitrogen, oxygen, or fluorine, involving dipole–dipole associations. Due to their unique directionality, adjustability, and specificity, hydrogen bonds are widely employed for cross-linking hydrogels. However, hydrogels formed solely through hydrogen bonding tend to exhibit poor mechanical strength and are prone to breakage. To enhance the mechanical strength, it is effective to introduce multiple hydrogen bonds or incorporate other cross-linking methods [73].

For instance, Jian et al. [86] designed a versatile adhesive hydrogel dressing with strong bioactivity and biocompatibility. This dressing consisted of two parts, adhesive and non-adhesive. In the non-adhesive component, methacrylate gelatin (GelMA), carrying bioactive compounds and exceptional biocompatibility, was encapsulated in a *N*-[Tris (hydroxymethyl) methyl] acrylamide (THMA) hydrogel matrix. This THMA-based hydrogel, which is abundant in hydroxyl and amino groups, contributes to dressing flexibility and strong adhesive capabilities. Dressing facilitates wound closure in mobile areas through hydrogen bonding and the energy dissipation effect, facilitated by hydroxyl and amino groups in THMA-based hydrogels. In addition, it creates a physical barrier for wound healing and effectively prevents external invasion and bacterial penetration.

Zhao et al. [87] engineered dual physical hydrogels utilizing quadruple hydrogen bonding and catechol-Fe^3+^ coordination. These hydrogels comprised cross-linked ureidopyrimidinone-modified gelatin with both catechol-Fe^3+^ coordination and quadruple hydrogen bonding. These versatile dressings displayed remarkable tissue adhesion, antioxidant capabilities, and responsiveness to both near-infrared (NIR) stimuli and pH levels, surpassing the performance of medical glue and surgical sutures in achieving superior wound closure.

Self-assembly refers to the spontaneous formation of an organized structure from fundamental structural units via non-covalent bonding interactions. For instance, amphiphilic polymers consist of hydrophilic and hydrophobic segments within each chain. These distinct segments have the ability to self-assemble in water or oil phases, forming hydrogels through interactions between their hydrophilic and hydrophobic components. Wang et al. [88] produced porous hydrogels through the self-assembly of the amphipathic amino acid peptide RADA16. These hydrogels facilitated the regeneration of skin appendages and exhibited potential applications in stem cell transplantation and tissue engineering.

### 11.2. Chemical Cross-Linking

Chemical cross-linking is a procedure in which chemical bonds are formed to connect polymer chains, forming a complex network structure under the influence of various factors such as light, heat, radiation, ultrasonic waves, and cross-linking agents. It encompasses diverse reactions such as Michael addition, Schiff base formation, free radical polymerization, click chemistry, and enzyme-mediated cross-linking.

Michael addition reaction: Wie et al. [88] drew inspiration from mussel adhesive proteins (MAP) and devised an injectable adhesive hydrogel based on dopamine (DOPA) for wound healing. This innovative dressing could be conveniently replaced using a zinc ion spray. They synthesized a highly branched polymer, HB-PBAE, through a Michael addition reaction involving dopamine (DOPA), poly(ethylene glycol) diacrylate (PEGDA700), and pentaerythritol triacrylate (PETA). The DOPA-based adhesive hydrogel was formulated by blending HB-PBAE, poly(1-vinylimidazole) (PVI), and gelatin solution, followed by the addition of Fe^3+^. Intriguingly, the application of a Zn^2+^ solution resulted in a rapid reduction in adhesive strength followed by a significant increase. Notably, this adhesive dressing minimized damage during dressing changes in the wound bed of mice and expedited the wound healing process.

Schiff base reaction: The Schiff base reaction involves the swift creation of a reversible imine bond between amino and aldehyde groups. Hydrogels can be created within minutes via this method and exhibit self-healing properties due to imine bonds being reversible; for instance, Giano et al. [89] utilized poly-dextran aldehyde (PDA) and polyethyleneimine (PEI) to fabricate a hydrogel through Schiff base cross-linking. This hydrogel exhibited antibacterial properties and demonstrated tissue adhesion behavior.

Free radical polymerization: Free radical polymerization consists of chain initiation, chain growth, chain termination, and chain transfer stages. Polymer chains containing terminal olefinic bonds are initiated and polymerized to form a network structure under conditions involving initiators, light, heat, radiation, and more. Photoinitiated radical polymerization offers precise control over the timing and location of cross-linking reactions, with applications that are highly convenient. However, the commonly used UV light source for photocrosslinking can negatively affect human tissue cells and pose health risks. As a result, investigation has focused on sources of light in the long-wavelength blue spectrum, infrared light, and visible light. For example, Hu et al. [90] fabricated a double-crosslinked o-nitrosobenzaldehyde-modified gelatin (GelNB)/chitosan methacrylate (CSMA) chitosan hydrogel using blue light-induced photocrosslinking and Schiff base reaction, demonstrating good biocompatibility, which could promote the healing of full-thickness wounds [90].

Click chemistry: Click chemistry encompasses various reactions, including the Diels–Alder reaction, tetrazine–norbornene chemistry, and the catalytic azide–alkyne cycloaddition reaction. These reactions are known for their mild conditions, high reactivity, and high yield, making them a focus of interest in hydrogel material development. For instance, the Diels–Alder reaction encompasses the creation of numerous ring structures between electron-rich conjugated dienes and electron-deficient double (or triple) bond compounds (dienophiles). Zhou et al. [91] employed this reaction to combine antibacterial imidazolium polyionic liquid into hydrogel networks for wound healing applications.

## 12. Wound Healing Hydrogels Based on the Needs of the Skin Healing Process

### 12.1. Hemostasis

In the initial stage of skin wound healing, the ability to quickly control bleeding is of great importance. Excessive bleeding is the leading preventable cause of death among injured military personnel and civilians [92]. Hydrogel materials with exceptional hemostatic properties play a crucial role in reducing the time required for wound healing. Approaches to boost the hemostatic properties of hydrogels include the following: (1) enhancing the materials’ adherence to tissues for efficient sealing to achieve hemostasis; and (2) enhancing their capacity to attract negatively charged platelets, thus facilitating coagulation processes [93].

Wound moisture is one of the obstacles to sticking the hydrogel on the skin. Diverse techniques have been utilized to facilitate adhesion to the commonly moist surfaces of skin wounds, encompassing bionic adhesion, hydrogen bonding, electrostatic interaction, dynamic covalent bonding, topological adhesion, and others. Interactions based on catechol have proven effective in creating adhesives well-suited for wet conditions [94]. In Liang et al. [95], dopamine (DA) was incorporated into hyaluronic acid, and a hydrogel was created by combining it with reduced graphene oxide (rGO) through the catalytic oxidation of HRP/H_2_O_2_. The hydrogel exhibited robust tissue adhesion owing to the various interactions between catechol groups and soft skin tissues. The HA-DA/rGO hydrogel, recognized for its formidable tissue adhesion, was regarded as a potential hemostatic agent.

Guo et al. [96] devised and manufactured an injectable hydrogel comprising quaternary ammonium chitosan (QCS) and tannic acid (TA), utilizing a straightforward mixing process of the two components under normal body conditions. This hydrogel primarily relied on dynamic ionic bonds and hydrogen bonds between QCS and TA, granting it remarkable injectable, self-healing, and adhesive properties. Leveraging the inherent antioxidant, antibacterial, and homeostatic capabilities of TA and QCS, this hydrogel demonstrated the ability to restrain reactive oxygen species, exhibit broad-spectrum antibacterial effects, promptly aid in homeostasis, and notably, enhance the healing of accelerated full-thickness wounds. Preman et al. [97] introduced a hemostatic small molecule, tannic acid (TA), into a hydrogel with pH-temperature dual responsiveness, consisting of sodium alginate and poly(N-vinyl caprolactam). The released TA was capable of reacting with blood proteins, resulting in immediate coagulation and providing an excellent hemostatic effect for this hydrogel.

To reduce blood loss and enhance platelet binding, Liu et al. [98] added ethylenediamine to carboxymethyl chitosan and created an aldohydroxyethyl starch/amino carboxymethyl chitosan (AHES/ACC) hydrogel using Schiff base reactions. The injectable AHES/ACC hydrogel acted as a physical filler, quickly minimizing blood loss. Additionally, the cationic amino groups interacted with anionically charged platelets, accelerating the blood clotting process. Therefore, AHES/ACC hydrogel could serve as an effective alternative to traditional medical gauze for tissue adhesion and hemostatic wound dressings.

Hydrogel dressings with a porous structure and rough surface can enhance their contact area with blood, promoting swift blood absorption and expedited hemostasis. A variety of synthetic and natural compounds have been utilized to create the optimal hemostatic agent. Sodium alginate (SA) has gained popularity in the synthesis of hemostatic materials due to its excellent biocompatibility, biodegradability, high hygroscopicity, and ion-exchange properties. Huang et al. [99] developed a hemostatic composite based on alginate with remarkable hemostatic and high water absorption characteristics. Pan et al. [100] produced an sodium alginate/human-like collagen/poly(vinyl alcohol) composite hydrogel as a porogen. Upon contact with blood, the hydrogel dressings allowed blood to enter the internal porous structure, facilitating rapid absorption. Additionally, alginate helped enrich calcium ions in the wound blood, thus accelerating coagulation reactions to expedite hemostasis.

### 12.2. Antibacterial Properties

Infection is one of the most important factors after an injury, it can lead to a prolonged wound healing process and potentially more severe secondary wounds. Therefore, the research and development of wound repair materials that seamlessly match skin tissue and prevent bacterial invasion can significantly reduce the risk of infection. Various methods for preparing antibacterial hydrogels have been developed. These approaches involve integrating antibacterial agents like antibiotics, antibacterial drugs, and metal nanoparticles into the hydrogel. Additionally, antibacterial components are introduced into the hydrogel network through either physical or chemical cross-linking [101]. Zhao et al. [102] engineered a multifunctional hydrogel (COC hydrogel) with dual cross-linking utilizing quaternary chitosan, methacrylate anhydride-modified collagen, and oxidized dextran. This double-crosslinked structure enhanced the hydrogel’s stability without compromising the graft function of the Schiff base. Notably, silver ions underwent rapid conversion in situ into silver nanoparticles (AgNPs) during the COC hydrogel formation, effectively mitigating issues related to dispersion and aggregation. In vivo findings demonstrated that COC@AgNP hydrogel expedites the healing of complete skin defects by fostering anti-infective and anti-inflammatory responses, stimulating collagen buildup, and facilitating the regeneration.

Tian et al. [103] combined hyaluronic acid with an ethylenediaminetetraacetic acid (EDTA)-Fe^3+^ complex to create a hydrogel with antibacterial and self-healing attributes. Fe^3+^ functioned both as a physical crosslinking agent and a non-toxic antibacterial substance. The hydrogel degraded when exposed to bacterial-secreted hyaluronidase (HAase), releasing Fe^3+^ complexes. These complexes were absorbed by bacteria, reduced to Fe^2+^ by H_2_O_2_, and generated hydroxyl free radicals that destroyed bacterial proteins and nucleic acids. This sustained release of Fe^3+^ achieved highly effective antibacterial properties. In another study, Singh et al. [104] utilized 2-hydroxyethyl methacrylate (HEMA), gum arabic (GA), and carbomer to formulate hydrogels containing the antibacterial drug moxifloxacin. These moxifloxacin-loaded hydrogels resulted in reduced wound inflammation, along with enhanced collagen and capillary formation. The hydrogels acted as drug reservoirs, adjusting the diffusion rate of molecules for localized and sustained release based on the patients’ needs in the wound healing process.

The introduction of exogenous active substances into materials can confer antibacterial properties. Giano et al. [89] cross-linked polydextran aldehyde (PDA) and polyethyleneimine (PEI) through Schiff base bonds. The PEI material itself exhibited antibacterial properties due to the abundance of protonated amines in its structure at physiological pH. It could interact with negatively charged components to disrupt the cell wall and cell membrane of bacteria, leading to bacterial lysis. Quaternized chitosan (QCS) is a widely researched antibacterial polymer. In another study, Du et al. [105] used a mixture of QCS and polyethylene glycol diacrylate (PEGDA) to synthesize hydrogel under ultraviolet light. In vivo, the hydrogel demonstrated excellent absorption and degradation, resulting in seamless wound closure.

### 12.3. Anti-Inflammatory Properties

Inflammation is a critical component of the second stage of wound healing. Proper inflammation plays a vital role in the wound repair process. However, persistently high levels of pro-inflammatory chemokines and reactive oxygen species (ROS) concentrations can lead to nonhealing at the injury site. To address this issue, anti-inflammatory hydrogels have been developed to promote the polarization of macrophages, reduce pro-inflammatory chemokines, and eliminate ROS [84].

The most convenient way to enhance the anti-inflammatory properties of hydrogels is to introduce anti-inflammatory drugs. Lei et al. [106] created a hydrogel that acts as an antioxidant and anti-inflammatory agent. This hydrogel is made from a secondary starch polymer produced by the eye worm, which possesses the ability to clear reactive oxygen species (ROS), thereby facilitating the healing of wounds. Zhang et al. [107] formulated an intensified antioxidant hydrogel by merging antioxidants with the hydrogel matrix, aiming to amplify antioxidant effects and improve the efficiency of clearing ROS. Moreover, the incorporation of certain metal nanoparticles into hydrogels can aid in regulating the inflammatory response. When subjected to an external magnetic field, magnetic nanoparticles can govern macrophage polarization and manage the release of growth factors. This feature makes magnetically responsive hydrogels a promising choice for anti-inflammatory wound treatment. Nevertheless, it is essential to highlight that an excess of reactive oxygen species (ROS) can intensify the inflammatory response and hinder wound repair. Hence, maintaining an optimal ROS level is especially crucial for the healing of chronic wounds [107].

### 12.4. Antioxidant Properties

An excess of inflammation can result in a notable elevation of reactive oxygen species (ROS), and consequently, the buildup of ROS can further exacerbate the inflammation [108]. High levels of ROS not only harm cells and DNA but also hinder blood vessel regeneration, consequently slowing down the wound healing process. As a result, it is crucial to develop antioxidant hydrogels that can sustain a low concentration of reactive oxygen species (ROS) to promote the healing of wounds [109]. The design of most antioxidant hydrogels involves either directly incorporating antioxidants or grafting antioxidant molecules onto them. However, developing hydrogels using materials with inherent antioxidant properties is highly significant [84].

Natural antioxidants encompass various categories, including thiol compounds like GSH and γ-glutamyl-cysteinyl-glycine, non-thiol compounds such as polyphenols and anthocyanins, vitamins like ascorbic acid, alpha-tocopherol, and vitamin A, and a range of enzymes like catalase, GSH-reductase, and GSH-peroxidase. Within living organisms, in vivo antioxidants comprise both enzymatic (e.g., SOD, catalase, peroxidase, and glutathione peroxidase) and non-enzymatic varieties (e.g., vitamin E, vitamin C, nitric oxide, and metal-binding proteins). These antioxidants exhibit robust antioxidant capabilities, effectively combating free radicals within organisms to mitigate oxidative stress. Their mechanism involves either inhibiting the production or eliminating free radicals and impeding their chain reaction. Moreover, fostering the generation of non-enzymatic antioxidants or activating the body’s enzymatic antioxidant system can also hinder or delay molecule oxidation. Overall, the primary function of antioxidants is to aid the body in defending itself against damage caused by reactive oxygen species (ROS) [110,111].

Natural polyphenols, such as ferulic acid, tannic acid, anthocyanin, and others, are known for their potent antioxidant properties and are frequently used to confer antioxidant attributes to hydrogels. In Ahmadian et al. [112], utilizing the abundant hydrogen bonds between tannin and gelatin, an effective GelTA hydrogel for wound healing was developed. The GelTA hydrogel demonstrated various biological activities, including free radical scavenging, hemostasis, and antibacterial properties. To enhance the antioxidant and biomineralization characteristics of gelatin, Zhang et al. [113] synthesized dopamine-modified gelatin (Gel-DA). They then crafted a multifunctional Gel-DA@Ag/GG hydrogel by combining Gel-DA with guar gum (GG). Compared to unmodified gelatin (Gel) and GG, Gel-DA exhibited outstanding antioxidant activity.

Hydrogels modified with thioether also exhibit remarkable resistance to oxidation. Liu et al. [114] manufactured thioether-modified hyaluronic acid (HA) nanofibrous membranes through electrospinning and subsequently constructed a nanofibrous (FHHA-S/Fe) hydrogel using Fe^3+^ as a cross-linking agent. Due to the intrinsic properties of high molecular weight HA and thioether modification, the hydrogel not only facilitated the transformation of macrophages into the anti-inflammatory M2 phenotype but also displayed effective reactive oxygen species (ROS) scavenging capabilities. Manganese dioxide (MnO_2_) nanosheets have the ability to promote the decomposition of H_2_O_2_ to generate O_2_ and efficiently eliminate ROS. In addressing the issues of inadequate oxygen supply and high ROS levels in diabetic wounds, Wang et al. [115] incorporated MnO_2_ nanosheets into a polymer network to create a multifunctional antioxidant hydrogel. This hydrogel exhibited antioxidant and anti-inflammatory effects, significantly accelerating the wound healing process.

### 12.5. Angiogenesis

Promoting the development of new blood vessels in tissues is a significant focus in contemporary regenerative medicine. Blood vessels play a vital role in material exchange between the bloodstream and tissues. In the context of wound healing, the regeneration of blood vessels is essential for promoting nutrient transportation and oxygen exchange, which are crucial for skin repair. Fibrin, in particular, has the ability to promote the adhesion of endothelial cells (ECs) and induce vascularization at the injured site [116]. Chen et al. [117] used four-arm thiolated polyethylene glycol (SH-PEG) and silver nitrate (AgNO_3_) to create an injectable hydrogel through coordinate cross-linking. In the cross-linking procedure, the angiogenic drug deferoxamine (DFO) was incorporated. This hydrogel exhibited the capacity to stimulate angiogenesis. The DFO-loaded hydrogel showcased a more comprehensive vascular network structure and longer blood vessel length. Additionally, the addition of DFO augmented the antibacterial impact of Ag^+^.

Sun et al. [118] used polyethylene glycol diacrylate and dextran with amine groups and double bonds to create hydrogels under UV irradiation. Vascular endothelial growth factor receptor 2 (VEGFR2) was detected on the fifth day, and the early formation of vascular networks was observed. This hydrogel could stimulate the migration of endothelial cells to the wound site, expediting the formation of new blood vessels. Hsieh et al. [119] employed glycol-modified chitosan, benzaldehyde-terminated polyethylene glycol, and fi-brin to create a hydrogel (CF). This CF hydrogel was found to increase the number of vascular endothelial cells and generate additional vascular branch points.

Liu et al. [120] developed an injectable and self-healing hydrogel through the combination of chitosan (CS) and metal ions for the effective healing of infectious and diabetic wounds. Taking advantage of amino and hydroxy groups, CS molecular chains were connected to silver ions (Ag^+^) and copper ions (Cu^2+^). CS-Ag-Cu hydrogel was able to gradually deliver Ag^+^ (as an antibacterial agent) and Cu^2+^ (as an angiogenic agent) to the wound area over a long period of time. This hydrogel had good adhesion and water absorption ability, as well as good antibacterial and biocompatibility property for skin regeneration [120].

Beyond the hydrogel material itself or the incorporation of angiogenic drugs, hydrogels carrying growth factors such as vascular endothelial growth factor (VEGF) and basic fibroblast growth factor (bFGF) have demonstrated potential in promoting vascularization. Stem cells, such as adipose-derived mesenchymal stem cells (ADSCs) and bone marrow-derived mesenchymal stem cells (BMSCs), can undergo differentiation into epidermal cells and release various cytokines and growth factors, thereby contributing to blood vessel formation and aiding in wound repair [121]. For example, Wang et al. [122] utilized β-cyclodextrin (β-CD), dextran, and hyaluronic acid to formulate a hydrogel containing resveratrol (Res) and VEGF for burn wound healing. VEGF plays a role in inducing inflammatory cells to migrate to the injury site and promoting the proliferation and migration of endothelial cells, thereby supporting wound epithelial regeneration.

Tan et al. [123] engineered a hydrogel with a combination of thrombin and fibrinogen, cross-linked with calcium ions, into which bone marrow mesenchymal stem cells (BMSCs) were embedded. In addition, vascular endothelial growth factor (VEGF) was incorporated into the fibrin gel to stimulate the differentiation of BMSCs into endothelial cells. The results showed that this suitable hydrogel significantly increases the adhesion and proliferation of smooth muscle cells and endothelial cells (SMCs and ECs) in the direction of skin repair. Furthermore, hydrogels loaded with stem cells, such as mesenchymal stem cells (MSCs), have demonstrated the potential to enhance vascular tissue repair. Eke et al. [124] loaded adipose-derived stem cells (ADSCs) into a hyaluronic acid/gelatin hydrogel. The combination of ADSCs and hydrogels increased angiogenesis threefold compared to cell-free hydrogels. This approach proved effective in promoting blood vessel regeneration and accelerating wound healing.

## 13. Wound Care Products in the Market

As depicted in Figure 2, the wound healing market based on product resources is segmented into various categories.

By wound type: Categorization based on application includes chronic wounds (such as diabetes, pressure ulcers, venous leg ulcers, foot ulcers, and arterial ulcers) and acute wounds (divided into burns and trauma and surgical wounds). In 2019, the chronic wound segment generated the highest revenue and is expected to maintain this position, attributed to an increase in post-operative surgical wounds, a rise in the geriatric population, increased awareness and diagnosis, and technological advancements [125].

By product: Within this market category, products are classified into conventional wound care products and surgical wound care products. In 2020, advanced wound care products held the largest market share [126]. In recent times, advanced wound care products have progressively overtaken traditional wound care products in the market. These advanced products are becoming the primary choice for treatment and are expected to experience rapid growth due to the increasing prevalence of chronic diseases like diabetes and obesity, coupled with a growing demand for innovative wound care solutions [4]. Discharged products comprise foam dressings, hydrocolloids, alginates, and hydrogels [127]. Surgical care products encompass areas such as infection management, discharge management, active wound care, and medical equipment. Infection control products include dressings containing silver particles and those without silver particles [127].

By End User: This market segment is classified into hospitals and community health centers. In 2019, due to the rise in advanced surgical procedures, hospitals accounted for the largest share of the advanced wound care market. A broader range of patients preferred to visit hospitals for specific cases, including surgical wounds, burns, and other wound types. This category is anticipated to exhibit the highest growth rate owing to the availability of facilities and superior treatment options in the wound care sector [128].

By Geography: Geographically, the markets are segmented into regions, including North America, Europe, Asia, and Oceania. Developed countries, North America, and Europe dominated the international wound care market in 2020, with North America holding the largest market share [129]. The significant share in the regional market is attributed to the increasing prevalence of chronic wounds and surgical procedures. Some of the prominent products in the field of wound management in recent years are NUSHIELD, APLIGRAFT, Pura PLay AM, DERMAGRAFT, AFFINITY, AFFINITY, AMNIOFIX, GRAFIX and GRAFIX PL, and STRAVIX and STARVIX PL, as shown in Table 2.

## 14. Summary and Outlook

The process of healing skin wounds is intricate, and improper treatment methods can significantly impact this recovery. Consequently, there is a heightened focus on improving approaches for treating skin wounds. This article provides a summary of recent advancements in using hydrogels for such purposes. Hydrogel dressings, owing to their exceptional physical and chemical properties, stand out as an excellent alternative material for addressing minor surface injuries. Moreover, as a form of skin tissue engineering material, hydrogels show promise in substituting skin grafts for treating severe, deep skin injuries. They aid in the regeneration of vascular tissue and skin appendages.

While substantial research has delved into utilizing hydrogels for healing skin wounds, effectively treating chronic and deep wounds remains a persistent challenge. Future investigations into hydrogels aim to reduce costs while enhancing their properties. Addressing chronic wounds remains a primary focus for upcoming research endeavors. To achieve this objective, it is necessary to discover more bioactive hydrogel materials that can provide the following features:They must have advanced mechanical features so that they can withstand the dynamic nature of wound environments.Designing smart hydrogels can enable hydrogels to dynamically respond to changes in the wound environment and adapt properties based on specific healing stages.They must have advanced drug delivery capability and be effective with the controlled release of therapeutic agents to promote healing and prevent infections.The development of hydrogels with suitable degradation profiles can address concerns about their durability and possible removal methods.The hydrogel formulation must be tailored to the patient’s needs; considering factors such as skin type and wound characteristics can increase the formulation’s effectiveness and reduce side effects.

## Figures and Tables

**Figure 1 ijms-25-01982-f001:**
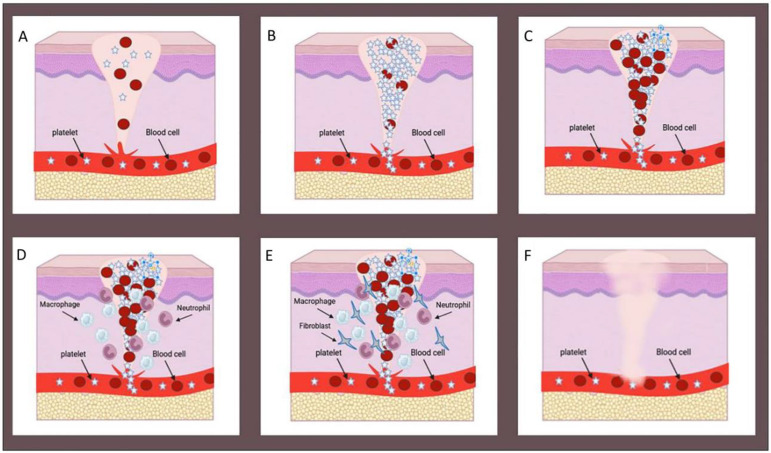
Schematic diagram of wound healing mechanism: (**A**) damaged blood vessel; (**B**) hemostasis platelet activation; (**C**) hemostasis fibrin clot; (**D**) inflammatory phase; (**E**) proliferation phase; (**F**) remodeling phase.

**Figure 2 ijms-25-01982-f002:**
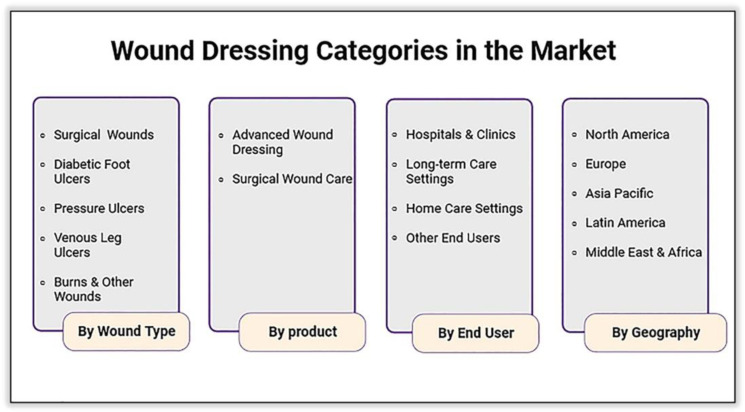
Categorization of the wound care market based on application.

**Table 2 ijms-25-01982-t002:** Products for advanced wound healing.

Product	Company	Components	Ref.
NUSHIELD	Organogenesis in Canton, MA, USA	Comprising amnion, chorion, and a spongy layer	[130]
APLIGRAFT	Organogenesis in Canton, MA, USA	Incorporating living keratinocytes and stem cells	[131]
Pura PLay AM	Organogenesis in Canton, MA, USA	A collagen sheet treated with 0.1% poly hexamethylene biguanide hydrochloride	[132]
DERMAGRAFT	Organogenesis in Canton, MA, USA	Featuring living fibroblasts seeded on a bioabsorbable scaffold	[133]
AFFINITY	Organogenesis in Canton, MA, USA	Utilizing fresh amniotic membrane as a wound covering	[134]
AMNIOFIX	MIMEDX in Marietta, GA, USA	Allograft consisting of amnion/chorion membrane	[135]
GRAFIX and GRAFIX PL	OSIRIS (a part of Smith and Nephew), Columbia, MD, USA	Placental membrane, inclusive of mesenchymal stem cells	[136]
STRAVIX and STARVIX PL	OSIRIS (a part of Smith and Nephew), Columbia, MD, USA	Umbilical tissue	[137]

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
