# Peer review of "Hydrogel-Based Skin Regeneration"

_ijms, 2024, doi:10.3390/ijms25041982_

Round 1
Reviewer 1 Report
Comments and Suggestions for Authors
The authors present a well written summary/review on Hydrogel for cutaneous wound healing. They report basic principles, and wound healing-related features.
Remaining issues before further consideration for acceptance are:
The authors should consider to shorten the manuscript from line 1-171. This is basic skin anatomy and wound healing knowledge from the text book. The two figures are also basic knowledge and can be omitted.
It would like to see materials in cross section, histologies, cell seeing pictures etc.
Furthermore, the authors should furnish an addition paragraph on published clinical randomized studies, Cochrane reviews, and metanalysis on current, commercially available hydrogels.
Author Response
|
Comments 1: The authors should consider shortening the manuscript from lines 1-171. This is basic skin anatomy and wound healing knowledge from the textbook. The two figures are also basic knowledge and can be omitted.
|
|
Response 1: In response to your recommendation, we have decided to remove this portion from the manuscript. We acknowledge your observation that the content is fundamental and aligns with textbook knowledge. Additionally, we have omitted the two figures associated with this section.
|
|
Comments 2: Furthermore, the authors should furnish an additional paragraph on published clinical randomized studies, Cochrane reviews, and metanalysis on current, commercially available hydrogels.
|
|
Response 2: Agree. We have, accordingly, modified to emphasize this point. In response to your recommendation, we have included an additional paragraph in our manuscript specifically focusing on current commercially available hydrogels. This addition enhances the depth of our discussion and provides a more comprehensive overview of the relevant literature.
|

Reviewer 2 Report
Comments and Suggestions for Authors
Comments:
1) Title: It should be revised as per the content of the manuscript. Manuscript mainly focused on wound healing related aspect, (Depth insight of skin regeneration has not been addressed appropriately).
2) Though the title based on ‘Hydrogel-based skin regeneration’, authors gave introduction about skin where various cell components were mentioned but skin resident stem cells, layers of epidermis etc. can also be added to make it more informative.
3) Figure 1 (line#105) and 2, quoted wrongly in the text. Text not matching the figure. Revise appropriately.
4) Use standard nomenclature and abbreviation (expand it when first used) throughout manuscript for e.g. b-1,4-linked D-glucuronic acid and (b-1,3) N-acetyl-D-glucosamine4?, Wang at al. (line#291)?, 108 CFU/mL (line #310), PEG (expand it when first used)
5) Major comment: Authors described various natural and synthetic polymers used in the hydrogel formation. Authors explained the basic properties of these polymers but the limitation of these polymers not addressed. Please discuss the limitations of each polymers and cite appropriate references in context of wound healing. Also, at the end, conclude the section with comparison of suitability of various polymers considering their advantages and limitations.
For e.g.
1) Collagen: Collagen type I, has many limitations i.e. it can trigger potential immune response, low mechanical strength and expensive. There are many references that suggested that properties of collagen are highly variable and dependent on a large number of fabrication parameters, such as collagen source or gelation pH etc. Therefore, it is highly recommended to discuss the limitations of each polymers by citing appropriate reference related to wound healing/skin regeneration.
2) Chitosan: Inadequate mechanical stability, uncontrolled biodegradation, bioaccumulation of degradation products and infection risks etc.
3) Hyaluronic acid (HA): HA is not able to form a physical gel alone, requiring chemical modifications, covalent crosslinking, and gelling agents to obtain a solid hydrogel.
6) A table can be incorporated summarizing the properties, advantage and limitations of various polymers
Comments on the Quality of English Language
Minor editing is required as suggested in the comments.
Author Response
|
3. Point-by-point response to Comments and Suggestions for Authors
Comments 1: Title: It should be revised as per the content of the manuscript. Manuscript mainly focused on wound healing related aspect, (Depth insight of skin regeneration has not been addressed appropriately).
Given the title of this article, " Hydrogel-based skin regeneration," the primary objective is to explore a key facet of wound dressing engineering. Specifically, the focus is on the critical process of selecting and combining the appropriate materials for hydrogel application, with the overarching goal of facilitating effective wound healing. Comments 2: Though the title based on ‘Hydrogel-based skin regeneration’, authors gave introduction about skin where various cell components were mentioned but skin resident stem cells, layers of epidermis etc. can also be added to make it more informative.
While we value the suggestion, it's important to note that the section on skin anatomy and histology has been excluded from the manuscript in response to a reviewer's request.
|
|
|
Comments 3: Figure 1 (line#105) and 2, quoted wrongly in the text. Text not matching the figure. Revise appropriately.
Thank you for your feedback. The suggested correction has been implemented.
Comments 4: Use standard nomenclature and abbreviation (expand it when first used) throughout manuscript for e.g. b-1,4-linked D-glucuronic acid and (b-1,3) N-acetyl-D-glucosamine4?, Wang at al. (line#291)?, 108 CFU/mL (line #310), PEG (expand it when first used)
I'd like to clarify that CFU (Colony Forming Unit) is indeed an abbreviation used as a unit of measurement for bacterial colonies. However, I appreciate your acknowledgment that the other points you raised have been addressed and corrected in the manuscript.
Comments 5: Authors described various natural and synthetic polymers used in the hydrogel formation. Authors explained the basic properties of these polymers but the limitation of these polymers not addressed. Please discuss the limitations of each polymers and cite appropriate references in context of wound healing. Also, at the end, conclude the section with comparison of suitability of various polymers considering their advantages and limitations.
The authors described a variety of natural and synthetic polymers used in hydrogel formation. The authors explained the basic properties of these polymers, but the limitations of these polymers were not considered. Please discuss the limitations of each polymer and cite appropriate sources for wound healing. Also, conclude the section by comparing the suitability of different polymers according to their advantages and limitations. Considering that this study was written to create a suitable resource to facilitate the comparison and selection of the right material for the treatment of wound dressings, the authors tried to state the most important advantages and disadvantages of the materials. However, according to your suggestion, Table 1 was added to the text.
Comments 5: A table can be incorporated summarizing the properties, advantage and limitations of various polymers
Thank you for your feedback. In response to your request, Table 1 has been incorporated into the manuscript.

Round 2
Reviewer 1 Report
Comments and Suggestions for Authors
Thank you for modifying the manuscript. It reads much better .